# Do extraordinary science and technology scientists balance their publishing and patenting activities?

**Yu-Wei Chang[1,2], Dar-Zen Chen[3], Mu-Hsuan Huang[1]***

1 Department of Library and Information Science, National Taiwan University, Taipei, Taiwan, 2 Center for Research in Econometric Theory and Applications, National Taiwan University, Taipei, Taiwan, 3 Department of Mechanical Engineering, National Taiwan University, Taipei, Taiwan

* mhhuang@ntu.edu.tw

**Data Availability Statement:** All relevant data are available from the Zenodo database. DOI: http://doi.org/10.5281/zenodo.5115346 Publication date: July 20, 2021 License (for files): Creative Commons Attribution 4.0 International.

## Abstract

This study investigated whether 12 scientists who had received the National Medal of Science and the National Medal of Technology and Innovation balanced publishing and patenting activities. The results demonstrated that although the scientist were recognized for their contributions to science and technology, the majority of recipients were not prolific researchers, and some were not influential. Notably, one scientist had not been granted a single patent. This indicated that scientific and technological contributions may not necessarily correspond with influential scientific publications and patents. Moreover, only two scientists had filed for patents before publishing, and they also invested more time developing technological inventions. Most recipients were science- or technology-oriented scientists. Few scientists balanced their publishing and patenting activities, and demonstrated excellent research and technology performance.

## Introduction

Scientists tend to demonstrate discoveries in research and invention by recording them in scientific publications and patents, respectively. Scientific publications and patents not only provide opportunities for scientists to establish their scientific and technological influence but also become a proxy through which researchers can explore scientific and technological activities and the relationship between them [1–6]. However, most scientists disseminate their discoveries through scientific publications, and not all scientists in fields related to technology contribute through inventions or are interested in producing patents [7]. Therefore, few scientists contribute both to the production of scientific publications and patents. Because scientists' research results have led to and followed technology and these researchers have produced scientific publications and patents to demonstrate their influence, we hypothesized that scientists engaging in both scientific and technological activities (hereinafter referred to as S&T scientists) produce both scientific publications and patents. Compared with research targeting scientists who only produce scientific publications, few studies have focused on S&T scientists [8–10], limiting our knowledge of their publishing and patenting activities.

**Funding:** This work was financially supported by the Center for Research in Econometric Theory and Applications (Grant no. 108L900204) from The Featured Areas Research Center Program within the framework of the Higher Education Sprout Project by the Ministry of Education (MOE) in Taiwan, and by the Ministry of Science and Technology (MOST), Taiwan, under Grant No. MOST 109-2634-F-002-045- and MOST 108-2410-H-002-219-. The funders had no role in study design, data collection and analysis, decision to publish, or preparation of the manuscript.

**Competing interests:** The authors have declared that no competing interests exist.

Considerable time and effort are involved in publishing and patenting. Nevertheless, scientists who produced both scientific publications and patents have been reported to have higher research performance than other scientists within the field [11, 12]. Scientists' research performance is mainly determined by productivity (number of publications) and influence (number of citations received by publications), which are quantified using bibliometric indicators. The scientific publication productivity of S&T scientists is typically higher than patent productivity because patenting requirements are more rigorous than those for publishing. However, the difference in productivity between publishing and patenting for individual S&T scientists remains unclear. Given the differences between disciplines and even across specializations within the same discipline [13], comparing scientists in terms of productivity was not the focus of this study. Instead, we investigated whether S&T scientists had a balanced ratio in terms of the number of papers published to patents awarded. Regarding the influence measured by citation counts, the skewness of citation counts received by scientific publications and patents [14] prompt another question regarding the difference in influence between scientific publications and patents for individual S&T scientists. In addition to indicators related to the quantity of scientific publications and patents as well as citation counts received by scientific publications and patents, the *h*-index, which combines the number of scientific publications and patents and their citation counts, has been widely used in the research evaluation context to elucidate scientists' research performance [14].

Considering the uneven distribution of scientific productivity among scientists, we focused on eminent scientists. Compared to normal scientists who accounted for majority of scientists, the relatively small number of eminent scientists are expected to have relatively smaller differences in productivity among them. Moreover, eminent scientists exhibit more prolific or higher influence than that of normal scientists [15, 16]. Therefore, eminent S&T scientists were assumed to have more scientific publications and patents for exploring the differences in research performance between publishing and patenting.

Bibliometric indicators have been controversial for identifying distinguished scientists. Therefore, to investigate the research performance of extraordinary S&T scientists, we identified scientists who have been granted two distinguished awards that separately emphasize contribution to science and technology. Considering the requirements close to eminent S&T scientists, finally, we targeted scientists who have received both the National Medal of Science (NMS) and the National Medal of Technology and Innovation (NMTI). As of 2019, only 12 individuals have been awarded both the NMS and NMTI. The low number of winners indicates that this select group has been responsible for extraordinary contributions to science and technology. Therefore, they are qualified to serve as representative research subjects because they demonstrate a great deal of influence in the fields of science and technology.

Because eminent S&T scientists were considered in this study, further hypotheses were proposed on the basis of our general impression of scientists with scientific and technological contributions [15–18]. In addition to producing both scientific publications and patents and having a higher scientific publication productivity than patent productivity, these excellent scientists have influential scientific and technological outputs. At least one scientific publication and one patent by each scientist has been highly cited by other scientists. If these contentions could be demonstrated to be true, that would indicate these scientists have produced scientific papers and patents and have demonstrate their contributions to science and technology. Moreover, they had a high number of scientific publications. A balanced ratio of publishing to patenting activity was evident among extraordinary science–technology scientists. Notably, the biggest difference between this study and previous studies is that individuals' publishing and patenting activities were covered. The relationship between publishing and patenting activities of the same scientist was monitored. We determined that individual scientists regarded as

having demonstrated a balance in publishing and patenting activities had a balanced ratio of the value of the *h*-index in scientific publications to the value of the *h*-index in patents. The *h*-index measuring both productivity and influence was used to reflect the difference in research performance between scientific and technological activities.

The 12 scientists considered in this study have been recognized for their excellent contributions to science and technology. Thus, we determined whether they balanced their publishing and patenting and also investigated the influence of their publishing and patenting activities. This study addressed the following three research questions:

- Do the 12 recipients with contributions to science and technology have similar ratios of scientific articles to patents?

- Are the scientific articles and patents produced by the 12 recipients highly cited?

- Do the 12 recipients have similar *h*-index ratios for scientific articles and patents?

## Literature review

### Scientists with scientific publications and patents

Among numerous channels through which scientists can disseminate their ideas, scientific publications and patents are common types of intellectual outputs. However, the majority of researchers publish their research results in only scientific publications [19]. Not all scientists can produce patents. Moreover, the field of academic research affects the cost and opportunities for researchers to be involved in industrial research and filing patents. Scientists within the same field demonstrated different levels of research output. For instance, materials science researchers conducting application-oriented research are in a better position to file patents than scientists conducting basic research [20]. This observation was consistent with the findings of Bonaccorsi and Thoma [11], who studied the research productivity of scientists in nano science and technology.

Although scientists can present the same concept in both a scientific paper and a patent, the emphasis placed on the value of science and technology affects the types of output and the approaches adopted for intellectual output [21]. Calderini et al. [20] found that patents filed by materials science researchers tend to be derived from their ideas published in journals with median and high impact factors. This study revealed an increase in scientific publications and a decrease in filing patents. Chang et al. [22] analyzed patent–paper pairs of scientific papers and U.S. patents in the field of fuel cells, and they determined that patents were filed and approved before the publication year of scientific papers within patent–paper pairs. The competitive relationship between science and technology implies that individual scientists do not exhibit the same productivity with respect to generating scientific papers and patents.

Higher research productivity and research influence were determined to be linked to researchers with scientific publications and patents in specific fields. Meyer [8] investigated differences in research performance between scientists with scientific publications and patents and scientists without patents in the fields of nanoscience and nanotechnology. Scientists producing scientific publications and patents demonstrated higher productivity and research influence than other scientists did. Bonaccorsi and Thoma [11] in a study on the research productivity in the field of nano science and technology demonstrated that two-thirds of U.S. patents were invented by all scientists with at least one scientific publication or a patent. Klitkou and Gulbrandsen [12] observed that researchers affiliated with universities in Norway who had published both scientific publications and patents in the field of life science had higher

research productivity than researchers without patents. These findings were consistent with claim by Magerman et al. [9] and Grimm and Jaenicke [10] that researchers involved in patenting would not reduce their research performance as measured by scientific publications.

## Influence and contribution

Research output is the principal method by which researchers demonstrate their influence and contribution. Scientific contribution was ranked as the second most important criterion in the evaluation of scientific papers, after research originality [23]. This indicates that the value of scientific papers can be demonstrated through their possible contribution to science. Similarly, patents reflect technological innovation and contribution. Scientometric researchers widely use citation counts as a proxy for scientific and technological influence [24–26] and also apply this bibliometric approach to assess the scientific contributions of individual scientists [27, 28]. This methodology indicates that research influence is not distinguished from scientific contribution. Although influence is not equivalent to contribution, the concept of influence is often mixed with that of contribution [29].

From the bibliometric perspective, the number of citations received by scientific publications is used to demonstrate various characteristics embedded in scientific publications, such as research quality and influence [30, 31]. However, opponents of the use of citation indicators for measuring research quality regard the limitations of citation indicators as barriers to quantifying the complex concept of research quality, which is mainly characterized by soundness, originality, scientific value, and societal value [31]. Although influence is regarded as having a closer relationship with citation counts than with research quality [32], complex citing behavior that involves diverse motives has resulted in some researchers being unconvinced that citation-related indicators are appropriate for assessing research influence [14, 30].

Several researchers have disputed that citation measures can be used to assess researchers' contributions to science [30]. Martin and Irvine [32] defined contribution to science as scientific progress and stated that the actual influence of a publication is highly associated with the concept of scientific progress. They also argued that papers with a substantial actual influence were those providing major contributions to scientific knowledge, and influence cannot be measured directly by citation counts. The bibliometric approach simplifies the notion of scientific influence and contribution, and thus peer review, a traditional approach, remains a common means for selecting recipients of awards and honors [33–35]. Traditionally, personal testimonials of the influence that a specific scientist has had on other researchers were the only manner in which to demonstrate scientific contributions. Despite the existence of peer review bias among reviewers, it is still considered the optimal method for assessing scientific contributions [23]. Nonetheless, some researchers support the citation count method; for instance, over half of the scientists interviewed for Aksnes's study [29] agreed that citation counts received by papers reflect their scientific contributions and value. However, we did not identify any awards that planned to incorporate citation counts into the award requirements.

Because of differences in the characteristics of the bibliometric and peer review approaches, numerous studies have attempted to examine the correlation between peer review and citation measures. Several studies have demonstrated a positive correlation between peer review and citation measures [36, 37], whereas other studies have reported a weak correlation [31, 38]. Baccini and Nicolao [31] observed the low degree of agreement for grading journal articles in 13 of 14 fields between the peer review and bibliometric methods. Abramo, D'Angelo, and Reale [39] did not claim that bibliometric indicators were superior to expert judgment, but they concluded that the bibliometric approach was more reliable than peer review for predicting the future scholarly influence of scientific publications. These findings indicate that

citation counts are a possible indicator of influence or contribution, and thus, the bibliometric approach was adopted as one indicator in the present study.

Altmetrics is a method for assessing the social effects of publications using social media or other Internet resources and can be employed for a more complete overview of the effect of publications. Therefore, it has become an area of interest; the increasing number of studies investigating the correlation between Altmetric scores and citation counts reflects a growing interest [40, 41]. Studies have reported a significant positive correlation between Altmetric scores and citation numbers [42, 43], whereas other studies have reported a weak correlation [44]. Altmetric scores should be regarded as a supplementary assessment measure of the influence of publications, but Altmetric data are only available for studies published after social media platforms were established in 2011 [45, 46]. Therefore, the public influence of studies published before 2011 cannot be assessed using Internet-based analysis. The majority of the scientific papers from the 12 subjects of the present study were published before 2011; therefore, we did not include Altmetric scores in our analysis.

## Methodology

### Scientists selection

Both the NMS and NMTI have been awarded to 12 researchers, who were selected as the subjects of this study. The NMS award was introduced in 1962 and the NMTI was introduced in 1985. The NMS and NMTI were established by the U.S. Congress in 1959 and 1980, respectively. The NMS is awarded to honor one scientist per year who has contributed to science and other areas and has influenced industries, education, or the country. The NMTI honors scientists who have greatly influenced and contributed to the American economy, social welfare, or the environment through technology commercialization and innovation. As of 2019, only 12 scientists have received both the NMS and NMTI. The NMS and NMTI candidates must be American citizens; therefore, these awards are not international in scope, but they are highly prestigious national awards. Thus, recipients of the NMS and the NMTI should constitute an appropriate sample for investigating the relationship between science and technology from the individual angle.

Table 1 lists the ages and institutions of the 12 scientists who received the NMS and NMTI. Subjects were aged 46–89 years when receiving the NMS and 53–85 years when receiving the NMTI. Most recipients first received the NMS and then the NMTI. Only three subjects received the NMS before the NMTI. The length of time between receipt of the two awards was 1–23 years. Seven scientists were awarded the two medals within 8 years of each other. The youngest subject was 70 years old in 2018 when data were collected for this study. This indicated that at the time of this study the 12 scientists were of retirement age and thus at the end of their academic career. Therefore, the scientific papers and patents by each recipient that were collected represented almost the entirety of their professional productivity.

### Data collection

Personal data including age, education, professional experience, and honors and awards were obtained from websites and social media. This assisted in identifying the papers and patents produced by the 12 recipients. The research papers and patents produced were the focus of this study and served as proxies for scientific and technological results, which are highly associated with professional output. Research papers published by each scientist were retrieved from the Scopus database, the largest multidisciplinary citation index database. The coverage of journals indexed by the Scopus database is much larger than Web of Science (WoS), another multidisciplinary citation index database. Therefore, to collect a more complete list of

**Table 1. Ages and institutions of the 12 scientists who received an NMS and NMTI.**

| No. | Name | Affiliation | NMS | Field (NMS) | Age (NMS) | NMTI | Field (NMTI) | Age (NMTI) |
|---|---|---|---|---|---|---|---|---|
| 1 | Robert S. Langer | Massachusetts Institute of Technology | 2006 | Engineering | 58 | 2011 | Medicine | 63 |
| 2 | Jan D. Achenbach | Northwestern University | 2005 | Engineering | 70 | 2003 | Aerospace | 68 |
| 3 | Herbert W. Boyer | University of California, San Francisco | 1990 | Biological Sciences | 54 | 1989 | Medicine | 53 |
| 4 | Nick Holonyak, Jr. | University of Illinois | 1990 | Engineering | 62 | 2002 | Electronics | 74 |
| 5 | Arnold O. Beckman | California Institute of Technology | 1989 | Physical Sciences | 89 | 1988 | Aerospace | 88 |
| 6 | Stanley N. Cohen | Stanford University | 1988 | Biological Sciences | 53 | 1989 | Medicine | 54 |
| 7 | Paul C. Lauterbur | University of Illinois | 1987 | Physical Sciences | 58 | 1988 | Medicine | 59 |
| 8 | Robert N. Noyce | Intel Corporation | 1979 | Engineering | 52 | 1987 | Computer Science | 60 |
| 9 | Carl Djerassi | Stanford University | 1973 | Physical Sciences | 50 | 1991 | Environment | 68 |
| 10 | Harold E. Edgerton | Massachusetts Institute of Technology | 1973 | Engineering | 70 | 1988 | Electronics | 85 |
| 11 | Jack St. Clair Kilby | Texas Instruments | 1969 | Engineering | 46 | 1990 | Hardware | 67 |
| 12 | Clarence L. Johnson | Lockheed Corporation | 1965 | Engineering | 55 | 1988 | Aerospace | 78 |

publications by each scientist over their course of career for analysis, the Scopus database was used as the source database to collect the published research papers. Only articles were defined as research papers in this study; nonresearch papers such as interviews, editorial materials, and book reviews were excluded. The latest year of publication was 2018. Regarding patents, because all 12 subjects were Americans and U.S. patent information was the only available data source, U.S. patents granted to the 12 subjects were retrieved from Google Patents. For purposes of this study, the latest year in which a U.S. patent was granted was also 2018. The names of the 12 scientists were the key information for searching bibliographic records of research papers and U.S. patents. Author and inventor affiliations were examined on the basis of their professional experiences to identify the relevant research output.

The bibliographic records of research articles provided by the Scopus database include title, author names, author affiliations, journal source, volume and number, pages, publication year, abstracts, and the number of citations of the paper. These data were downloaded on January 29, 2019. Because the cumulative number of citations received by each paper is updated to a date close to the end of 2018, the number of citations received by each paper closely reflects their influence as of the end of 2018. As mentioned in the preceding paragraph, Scopus covers more journal titles than WoS. This helped us collect a greater proportion of the articles published by the 12 scientists. However, Scopus does not provide the citation figures for years before 1970. For articles published before 1970 and that received citations before 1970, their annual number of citations for that period was collected from WoS if that citation data were available. The citation data collection was conducted on the premise that articles published before 1970 are indexed by WoS. Therefore, for articles published before 1970, their annual citation counts were collected from WoS (before 1970) and Scopus (after 1970). For example, the total number of citations received by an article published in 1968 was calculated from the number of citations recorded in 1969 in WoS and the number of citations recorded during 1970–2018 in Scopus. This prevented an underestimation of the number of citations received by articles published before 1970. Regarding the bibliographic records of patents, patent numbers, titles, abstracts, inventor names, and grant dates were collected in February 2019. Because the number of patent-related citations is not provided in the U.S. Patent and Trademark Office (USPTO) database, we calculated the number of citations received on the basis of the references for all patents. The list of references does not change after the patents are granted. Therefore, 2018 is the latest year for which the number of patent citations is recorded.

## Data processing

To improve precision, an intensive manual task was performed to examine the bibliographic records of papers and patents. First, the titles of papers that were categorized as articles were examined, followed by papers of fewer than five pages without references. Second, the full texts were examined and some were excluded because they were the incorrect type; for instance, interviews, lectures, or editorial letters.

Several bibliometric indicators were used to measure and compare the differences in productivity and influence as evident in articles and patents. Regarding research productivity, considering the differences in individual lengths of professional careers, we calculated the average productivity per year for each scientist. Researchers who favor collaborative research tend to have higher productivity. Therefore, apart from full counting, fractional counting was also used to calculate productivity. If an article was written by *n* authors, each author was granted one article and $1/n$ article for research productivity, respectively. Regarding indicators related to influence and as frequently done by researchers, citation counts were employed to represent influence and contribution over time [47]. The number of citations added per year was also tracked to assess variations in influence over time. Particular attention was paid to the most highly cited article and patent of each scientist to track influence over time. The subjects were all senior scientists that had made considerable contributions to their fields throughout their careers. Therefore, h-index that incorporated productivity and influence was used as a reference indicator of their overall research performance [48, 49].

## Results

### Contributions acknowledged by NMS and NMTI

Table 2 presents the scientific and technological contributions of the 12 scientists for which they were awarded the NMS and NMTI; data were obtained from the NMS and NMTI websites. Every recipient earned both their medals for the same or similar contributions. This indicates that an identical idea can be developed in scientific research or applied in technological

**Table 2. Contributions, backgrounds, and job experiences of the 12 recipients.**

| No. | Name | Contributions for earning awards | Education backgrounds | Job experiences |
|---|---|---|---|---|
| 1 | Langer | Polymeric controlled drug release systems | Chemical engineering | MIT |
| 2 | Achenbach | Wave propagation/ ultrasonic methods | Aeronautics and astronautics | Northwestern Univ. |
| 3 | Boyer | Recombinant-DNA technology | Biology; Chemistry; Biotechnology | Standard Univ.; Univ. of California; Genentech |
| 4 | Holonyak | Light-emitting diodes (LEDs) | Electronic engineering | General Electric Co.; Univ. of Illinois at Urbana-Champaign |
| 5 | Beckman | Development of analytical instrumentation | Chemical engineering; Physical chemistry; Photochemistry | California Institute of Technology; National Inking Appliance Co. |
| 6 | Cohen | Recombinant-DNA technology | Medicine | Standard Univ. |
| 7 | Lauterbur | Nuclear magnetic resonance | Chemistry | Univ. of New York at Stony Brook; Univ. of Illinois at Urbana-Champaign |
| 8 | Noyce | Integrated circuit | Physical electronics | Intel Corp.; Fairchild Semiconductor Corp. |
| 9 | Djerassi | Oral contraceptives / Insect control products | Chemistry | Syntex Co.; Standard Univ.; Wayne State Univ. |
| 10 | Edgerton | Stroboscopic photography | Electronic engineering | Edgerton, Germeshausen, and Grier, Inc.; MIT |
| 11 | Kilby | Integrated circuit | Electronic engineering | Texas Instruments |
| 12 | Johnson | Design of aircraft | Aeronautical engineering | Lockheed |

areas. For instance, Lauterbur received the NMS in the field of physical sciences in 1987 for nuclear magnetic resonance research, and he received the NMTI the following year in the field of medicine for his invention of nuclear magnetic resonance. Furthermore, Lauterbur's educational background in chemistry did not limit his scientific and technological contributions to his field. Seven recipients had engineering expertise. Moreover, scientists had educational backgrounds in medicine, chemistry, biology, or physics. Eight were practice-oriented academics. Two recipients did not work for industries and another two did not teach in universities.

## Awards

Fig 1 shows the number of awards, including the NMS and NMTI received by each scientist in specific years. In addition to the NMS and NMTI, each scientist earned several other scholarly awards during their professional career. The normal vertical line set as zero on the horizontal axis represents the year the NMS was awarded and the vertical bold red line represents the year the NMTI was awarded. Except for Edgerton, whose last award was the NMTI, all the scientists received other awards before and after both the NMS and NMTI. This means that their professional achievements and contributions were recognized several times by peers. The receipt of several distinguished awards reduces doubts regarding award committee bias. Except for Langer, who received up to 79 awards, the other 11 scientists received 8–26 awards. Their achievements in terms of the number of awards received may be attributed to the Matthew effect: that success breeds success [50]. Among 12 scientists, nine earned other major

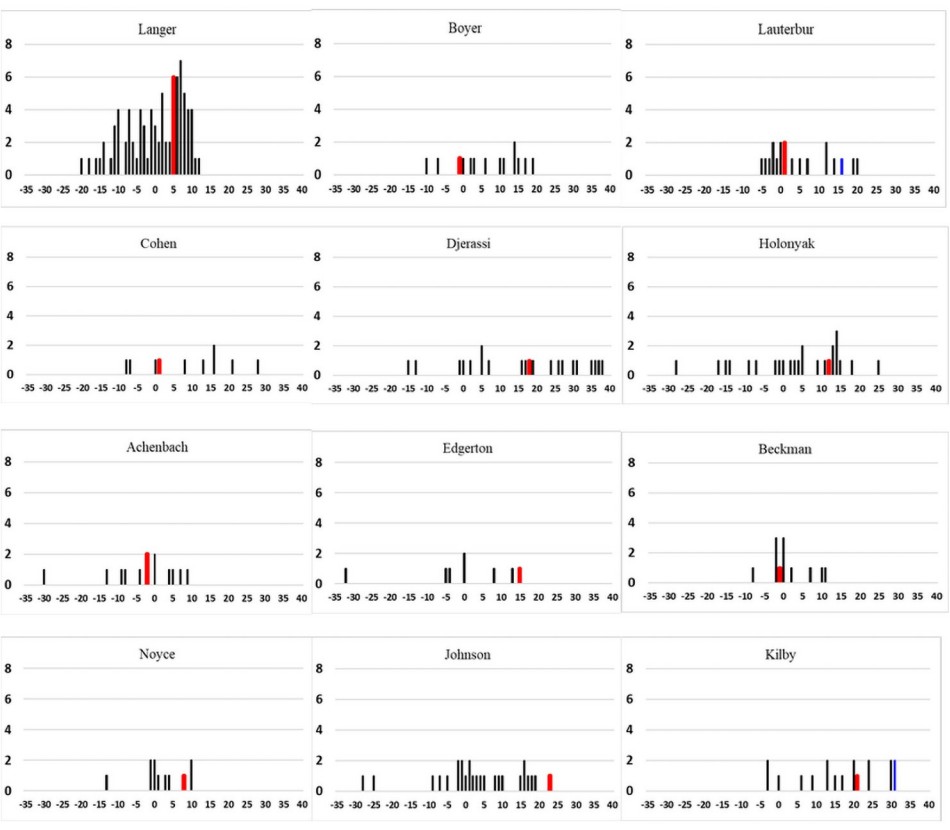

**Fig 1. Number of awards per year.**

awards and prizes in medicine, biotechnology, chemistry, and physics, such as the Albert Lasker Basic Medical Research Award, Dickson Prize in Medicine, Charles Stark Draper Prize for Engineering, Harvey Prize, Shaw Prize, and Wolf Prize; receipt of these awards was determined by checking the distinguished awards listed on the website of Harvard Medical School [51]. Two scientists were Nobel Laureates (Kilby and Lauterbur). The vertical blue line in Fig 1 marks the year that the researcher received the Nobel Prize. Kilby and Lauterbur received a relatively modest number of awards, at 18 and 19, respectively. Two scientists with a relatively low number of awards all specialized in aeronautics (Achenbach and Beckman), which is a relatively small field. Furthermore, Boyer and Noyce had built a close collaborative relationship because of their shared research interests. They received a similar number of awards.

## Research and technology productivity

Table 3 lists figures related to the productivity and influence of articles indexed by Scopus and patents granted by the USPTO for each recipient. Djerassi had the largest combined number of articles and patents. Excluding one recipient (Achenbach) who had not filed any patents, Boyer had the highest ratio of articles to patents (28.33), followed by Lauterbur (24.00). The higher the ratio is, the larger the difference between article productivity and patent productivity is. Scientists with more balanced productivity in research articles and patents were those

**Table 3. Ratios of articles to patents.**

| Name | A/P | A/P (F) | Time (A/P) | Ave Pro (A/P) | Ave C (A/P) | Ave HC (A/P) | H index (A/P) |
|---|---|---|---|---|---|---|---|
| Achenbach | - - | - - | - - | - - | - - | - - | - - |
|  | (377/0) | (184.3/0) | (56/0) | (6.73/0) | (22.9/0) | (10.2/0) | (51/0) |
| Boyer | 28.33 | 16.67 | 9.00 | 3.15 | 0.93 | 4.89 | 12.00 |
|  | (85/3) | (25/1.5) | (27/3) | (3.15/1) | (152.9/164.3) | (81.6/16.7) | (36/2) |
| Lauterbur | 24.00 | 32.50 | 13.33 | 1.92 | 6.73 | 33.97 | 10.50 |
|  | (100/4) | (45.5/1.4) | (40/3) | (2.5/1.3) | (87.5/13) | (42.9/1.3) | (42/4) |
| Cohen | 14.59 | 11.80 | 2.71 | 5.36 | 3.04 | 8.43 | 6.87 |
|  | (423/29) | (146.3/12.4) | (57/21) | (7.4/1.4) | (101.7/33.5) | (68.5/8.1) | (103/15) |
| Djerassi | 9.78 | 8.07 | 2.71 | 3.61 | 11.22 | 40.44 | 9.56 |
|  | (998/102) | (337.4/41.8) | (57/21) | (17.5/4.9) | (39.27/3.5) | (19.6/0.5) | (86/9) |
| Holonyak | 8.80 | 4.46 | 1.90 | 4.64 | 0.89 | 2.45 | 1.96 |
|  | (537/61) | (125.2/28.1) | (59/31) | (9.1/1.9) | (26.3/29.7) | (9.1/3.7) | (53/27) |
| Langer | 2.44 | 2.08 | 1.19 | 2.06 | 1.49 | 8.45 | 1.76 |
|  | (1171/479) | (255.1/108.3) | (44/37) | (26.6/12.9) | (123.3/82.7) | (297.6/35.2) | (194/110) |
| Edgerton | 1.70 | 1.03 | 2.20 | 0.77 | 0.86 | 1.00 | 1.30 |
|  | (56/33) | (31.6/30.8) | (33/15) | (1.7/2.2) | (8.3/9.6) | (0.6/0.6) | (13/10) |
| Beckman | 1.00 | 0.48 | 1.22 | 0.81 | 2.36 | 1.80 | 1.29 |
|  | (14/14) | (5.4/11.2) | (11/9) | (1.3/1.6) | (22.4/9.5) | (0.9/0.5) | (9/7) |
| Noyce | 0.35 | 0.14 | 1.00 | 0.36 | 12.98 | 10.61 | 0.50 |
|  | (6/17) | (2.1/14.8) | (6/6) | (1/2.8) | (320.8/24.7) | (29.6/2.8) | (5/10) |
| Johnson | 0.27 | 0.10 | 0.33 | 0.86 | 3.09 | 5.21 | 0.40 |
|  | (7/26) | (2.5/24.2) | (6/18) | (1.2/1.4) | (26.6/8.6) | (2.3/0.4) | (4/10) |
| Kilby | 0.03 | 0.04 | 0.08 | 0.45 | 2.18 | 0.74 | 0.05 |
|  | (2/58) | (1.5/37.7) | (2/26) | (1/2.2) | (64.0/29.4) | (3.0/4.0) | (1/21) |

*Notes*: A refers to articles; P refers to patents; Time refers to the cumulative years publishing articles/ producing patents; Ave proc. refers to the mean articles/patents per year. Ave C refers to the mean citation counts per article/patent; Ave HC refers to the mean citations received by the most highly cited article/patent per year.

with ratios of approximately 1. Only four scientists had an article-to-patent ratio equal to or below 1, which indicated that they were not typical academics who dedicated themselves to producing scientific articles; this is because having patents granted is more challenging than having articles published. When using fractional counting, Djerassi still had the highest total productivity in articles and patents. Regarding the ratios of article production to patent production, Lauterbur came in first place (32.50) and Boyer ranked second (16.67). Only the ranks of the top two scientists changed when two different methods were used for counting total productivity. Although no threshold was set for high productivity, the differences in individual productivity of articles and patents and average productivity per year signaled that the majority were not prolific authors and inventors. Moreover, we observed a substantial difference in productivity among the scientists. To account for the difference in productivity between disciplines, two scientists in the field of biomedicine who frequently collaborated, Boyer and Cohen, were taken as examples. Cohen produced a much larger number of articles (423 vs. 85) and patents (29 vs. 3) than Boyer produced.

Notably, the total numbers of articles and patents for each scientist represented their cumulative productivity over their careers, which exceeded the normal retirement age (65 years). We were surprised by the high productivity in both articles and patents produced by Langer and by the low productivity in articles and patents produced by Beckman, Noyce, and Johnson. To assess productivity over time, the average productivity per year was examined for each scientist. Five scientists published over five articles per year and five scientists were granted at least two patents per year. Only two scientists achieved high mean productivity in articles and patents. Furthermore, five scientists with a low ratio ($\leq 1.00$) were focused on the production of inventions instead of articles.

Excluding one scientist without any U.S. patents, eight scientists continued publishing and filing patents for over 30 years. However, the actual cumulative number of years for publishing articles and being granted patents (Time [A/P]) was found to vary between subjects (Table 3). The ratios of the Time (A/P) revealed that only three scientists had invested a longer time period to filing patents, with a range between 0 and 0.7. This implies the majority of scientists contributed more time to publishing. However, the ratios displayed may underestimate the time spent by winners, considering that filing a patent is more difficult than publishing an article because of the difference in the processes of filing patents and publishing articles. As expected, the majority of recipients had higher productivity in articles than in patents. Moreover, the findings demonstrated that only one scientist published articles after he had been a patent holder for 26 years. Another subject published his first article and was granted his first U.S. patent in the same year.

Fig 2 illustrates the number of articles and patents produced in specific years. Variations in productivity each year demonstrate each subject's productivity pattern. Moreover, the starting year indicates when individual scientists were first involved in publishing or patenting. The normal vertical line in the zero on the horizontal axis refers to the year they received the NMS, and the vertical bold line indicates the year when they received the NMTI. If a specific scientist received the NMTI five years after receiving the NMS, the year for obtaining the NMTI was labeled 5. Another reference point was set with a vertical line topped with a round dot, which refers to the year when a scientist was 40 years old. Regarding the horizontal lines, the black normal line represents the number of articles per year and the dotted red line refers to the annual number of U.S. patents. Fig 2 also exhibits the duration of time invested in publishing and patenting for each scientist. Some scientists started publishing articles before being granted patents or invested more time producing articles and thus obtained a higher number of articles. Article-oriented scientists are Achebach, Boyer, Holonyak, Lauterbur, and Cohen. Some scientists were patent-oriented and thus spent more time on invention; they received a

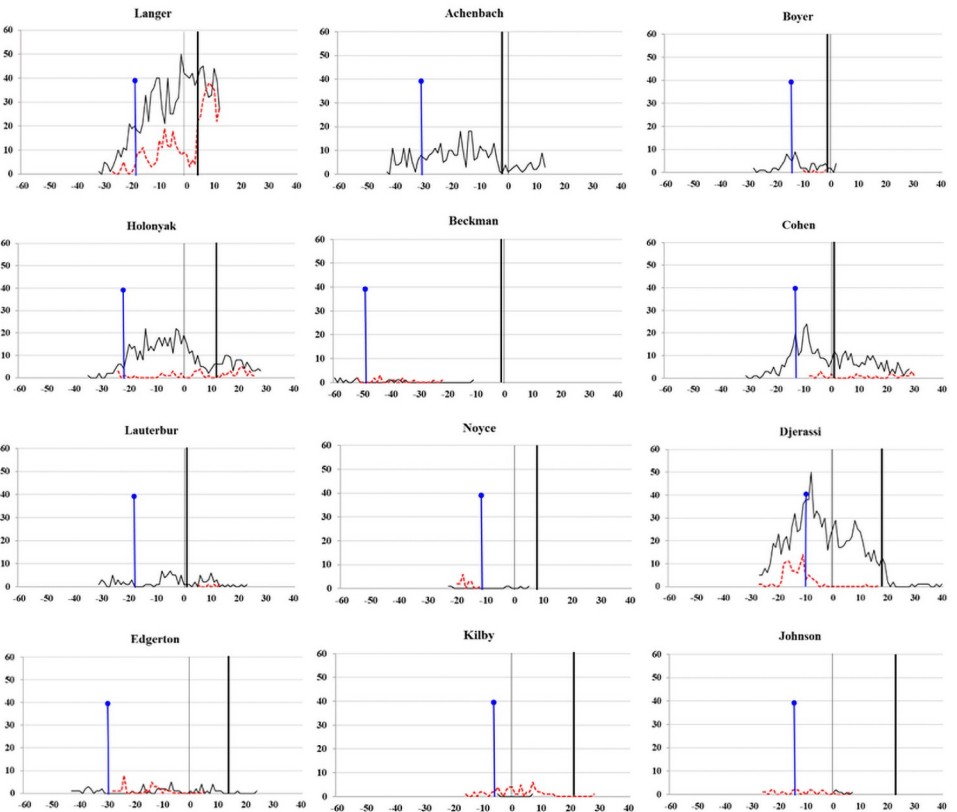

**Fig 2. Numbers of articles and patents by year for individual recipients.**

higher number of patents. Kilby and Johnson are such scientists. The remaining scientists were between being article-oriented and patent-oriented.

## Academic and technological influence

Mean number of citations was used as an indicator of the influence of articles and patents. The mean number of citations per article or patent is presented in Table 3. The ratio of the mean number of citations per article to the mean number of citations per patent indicates that Noyce (12.98) had the largest research influence despite only publishing six articles. Edgerton achieved the greatest mean technological influence with the lowest ratio of mean citations per article/patent (Ave C [A/P]) (0.86). Although Holonyak and Edgerton produced a lower number of patents than articles, their average technological influence per patent was higher than their average science influence per article. The influence of scientists may not be evident in the mean number of citations per article or patent because of the large variation in citation counts for individual articles and patents. Therefore, we further investigated the articles and patents with the highest citation counts. Regarding the average number of citations per year, a larger range of ratios of the most cited article to the most cited patent (AveHC [A/P]) was observed (0.74–40.44) than the range for AveC (A/P) (0.86–12.98).

To estimate the level of influence of the most-cited article for each scientist, we assessed their most-cited article in their field and compared the citation level to general articles in the subject field published in journals covered by WoS. For instance, for one subject, the article with the highest number of citations was published in a chemistry journal. Therefore, we

examined all articles published in chemistry journals covered by WoS. In terms of citation counts, each of the 10 scientists' most highly cited article was ranked within the top 0.1% of highly cited articles. We considered the top 0.1% of articles with the highest number of citations to be the threshold for highly cited articles because a study by Rodriguez-Navarro 2011) on the citation counts of papers by Nobel Laureates reported that high levels of citations are a useful indicator to identify influential articles. We observed that 10 of the 12 scientists published at least one highly influential article in journals belonging to their fields.

Fig 3 illustrates the changes in the cumulative number of citations received by all articles and patents per year and the changes in the annual number of citations received by the article and patent with the largest number of citations. The normal black line refers to the average number of citations received by all articles in a specific year, and the bold solid black line represents the average citations received by all patents in a specific year. Because of the large difference in the number of citations received for articles and patents, the number of citations was transformed in the scale of base 10 logarithm to reduce wide-ranging quantities to small scopes. This allowed us to plot the changes in the influence of articles and patents per year in the same subfigure. Regarding the article and patent with the largest citation counts, the dotted blue line reflects the number of citations received by highly cited articles per year, and the bold

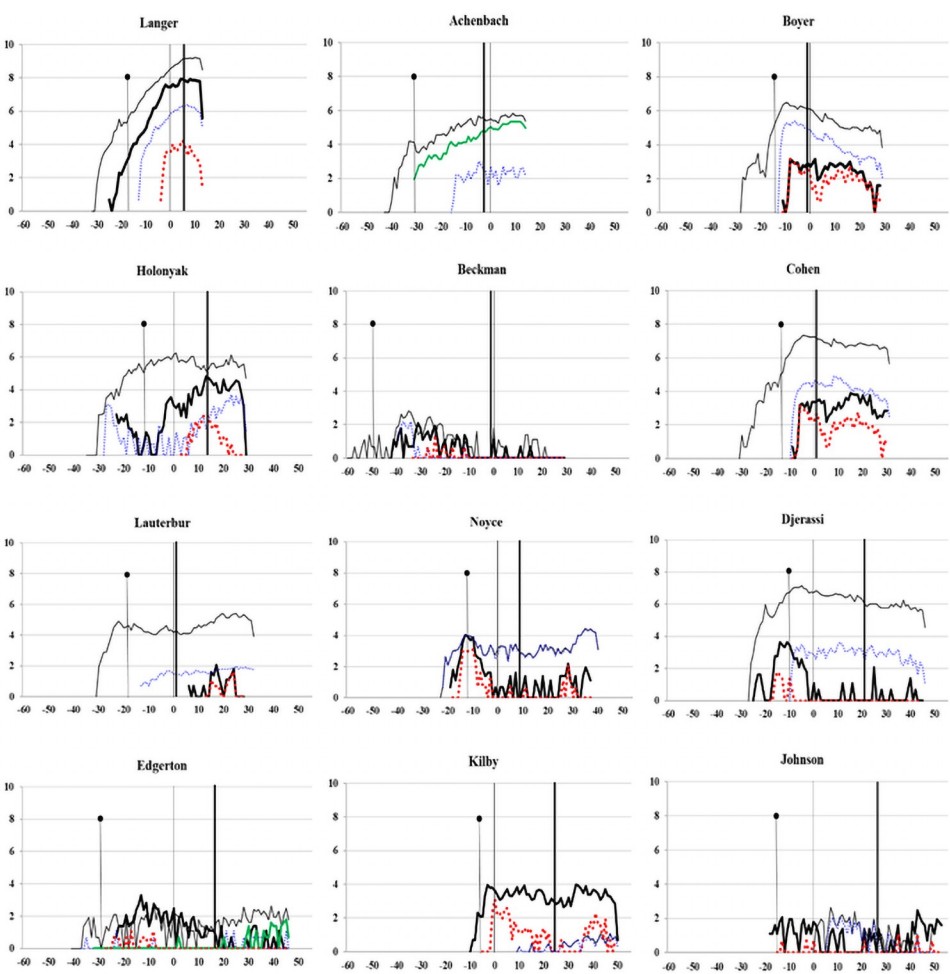

**Fig 3. Changes in the number of citations by year and type.**

dotted red line reflects the number of citations received by the highly cited patent per year. Because some scientists only published a limited number of articles with a low number of citations, we further searched whether they had published other types of scientific publications that had higher citation counts. We determined that two scientists (Achenbach and Edgerton) published books that have received a larger number of citations than their articles, according to the citation records of WoS and Scopus. Variations in the number of citations received by the most highly cited book per year were plotted with a green line. Excluding the scientist (Achenbach) who had not filed any patents, three of the remaining scientists had a greater influence from articles than from patents.

The *h*-index combines time, productivity, and influence and was used to reflect the longitudinal research performance of senior scientists. As displayed in Table 3, the ratios in the *h*-index of articles to patents demonstrated that Boyer had the highest *h*-index of articles and patents. Higher ratios appear to be related to larger differences in the production of articles and patents. Table 4 presents the correlations between pairs of indicators. A significant difference was observed in each pair of indicators related to the *h*-index (A/P), A/P, A/P(F), and Time (A/P) with strong correlations (correlation coefficients > 0.8). A medium correlation was observed between the *h*-index (A/P) and AveHC (A/P), with a correlation coefficient of 0.636. Fig 4 illustrates the three ratios of articles to patents by scientist and the ages at which each scientist began publishing and patenting. Each scientist was assigned an exclusive color to label their scores for five characteristics. The 12 scientists' data are provided in descending order according to the *h*-index (A/P) value. The exception color is gray-blue, which is used for two scientists sharing the same Time(A/P) value. Moreover, one scientist (Achenbach) did not produce patents. Therefore, for this scientist, the ratios related to articles to patents could not be calculated; thus, no association among the three ratios and ages of publishing the first article is presented in Fig 4. Fig 4 illustrates that the majority of these scientists invested more time in publishing and yielded a higher ratio of h-index(articles) to h-index(patents). Moreover, most scientists started publishing before they started obtaining patents. Before the age of 30, ten scientists had published their first scientific papers, whereas only two scientists had obtained their first patents. The differences in ratios of articles to patents, including A/P, Time(A/P), and h-index(A/P), revealed that the ratio of patenting to publishing was heavily skewed among these 12 scientists.

A considerable gap in the 12 scientists' productivity and the influence of articles and patents was observed over a long-term period covering the end of their professional careers. Because the *h*-index that is suitable for measuring the productivity of senior scientists and the influence

**Table 4. Correlations between pairs of indicators.**

|  | h-index (A/P) | A/P | A/P(F) | Time (A/P) | AveC (A/P) | AveHC (A/P) | AvePro (A/P) |
|---|---|---|---|---|---|---|---|
| H index (A/P) | 1 | 0.926** | 0.842** | 0.813** | 0.172 | 0.636* | 0.542 |
| A/P |  | 1 | 0.891** | 0.889** | -0.039 | 0.397 | 0.549 |
| A/P(F) |  |  | 1 | 0.961** | 0.117 | 0.594 | 0.361 |
| Time (A/P) |  |  |  | 1 | 0.062 | 0.497 | 0.204 |
| AveC (A/P) |  |  |  |  | 1 | 0.680* | -0.124 |
| AveHC (A/P) |  |  |  |  |  | 1 | 0.222 |
| AvePro (A/P) |  |  |  |  |  |  | 1 |

Notes:

** $p < .01$,

* $p < .05$.

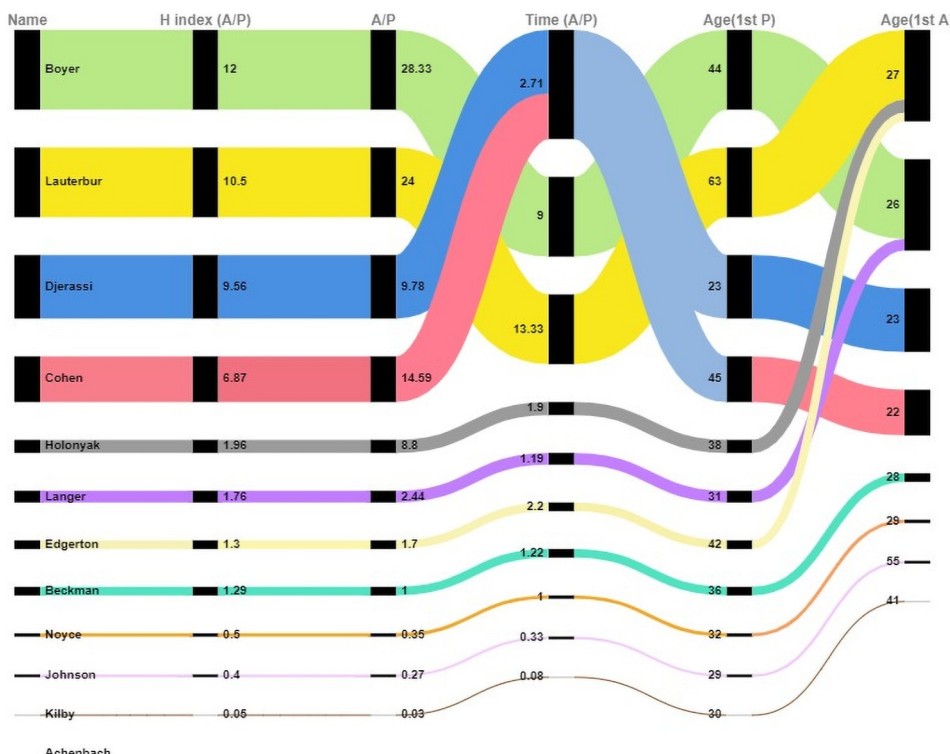

**Fig 4. Differences in three ratios of articles to patents by scientist and the ages at which each scientist began publishing and patenting.**

of the articles and patents that they produced, we used the results from the *h*-index for articles and patents to represent the differences between scientists in Fig 5 and divided them into three types according to their position in Fig 5. Only one scientist (Langer) was categorized as type A, featuring a high value of *h*-index in both articles and patents. A substantial difference in the value of the *h*-index of articles and patents was observed between Langer and other scientists. Although 11 scientists were concentrated on the left side, three scientists (Cohen, Djerassi, and Holonyak) who are closest to the center of Fig 5 were classified as type B because they were separated from the other eight scientists on the bottom left corner. The remaining eight scientists were classified as type C. According to the positions of the three scientists classified as type B, the requirements for separating type-B scientists from those of type C were generated. Three type-B scientists were located in the zone with an *h*-index score for articles larger than 50% and an *h*-index score for patents larger than 10%.

## Discussion and conclusions

The 12 subjects investigated in this study are the only scientists to have received both the NMS, which emphasizes contribution to science, and NMTI, which emphasizes contribution to technology. Receiving both these prestigious awards is a rare occurrence. These 12 recipients have proven their extraordinary professional achievements and contributions to science and technology. Therefore, we are not surprised that they received other noteworthy awards before and after the NMS and NMTI as evidence of their peers' recognition of their professional achievements. Eminent scientists can begin receiving honors early in their careers [52].

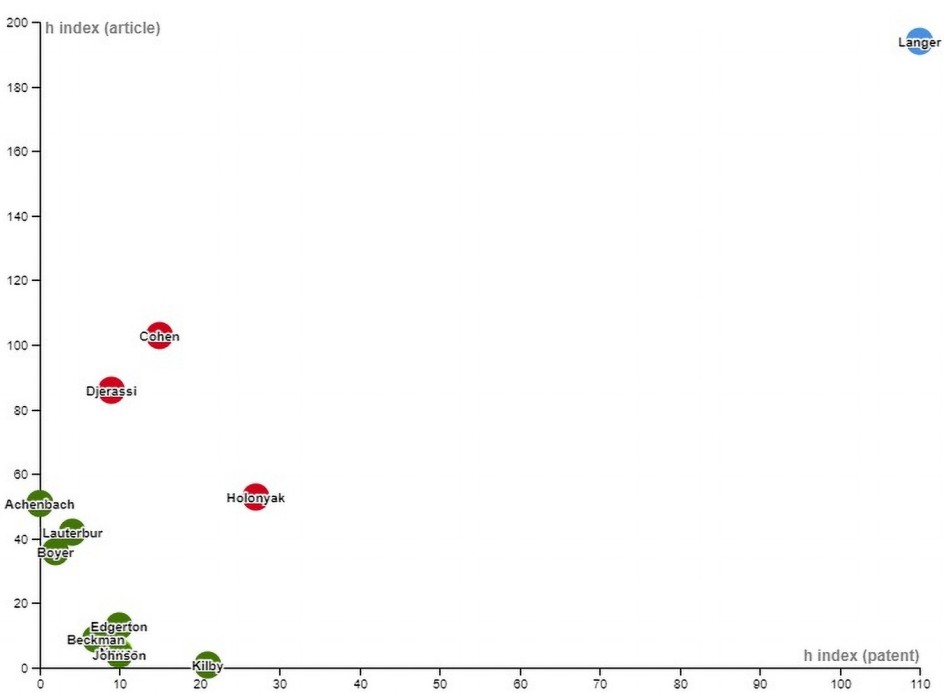

**Fig 5. Distribution of values of the article and patent h-index for each scientist.**

However, the 12 recipients' numerous distinguished awards along their professional careers originally elevated our expectation of observing their high research performance. Although we had no initial estimation of the number of articles and patents they produced and the number of citations received by their articles and patents, the 12 eminent scientists were expected to have a considerable number of articles and patents and have influential articles and patents. Given that research performance comparison between scientists in different fields is discouraged, this study focused on the differences in research performance in articles and patents for each scientist. We assessed whether eminent scientists balanced their scientific and technological activities and demonstrated a similar influence in these two activities by using the ratio-related indicators to compare article and patent productivity and their citation counts. Moreover, changes in productivity and influence by year were tracked along these scientists whole professional careers for obtaining more details regarding their scientific and technological outputs.

Although we observed an effect from the length of time on the amount of research output and citations received by papers and patents and subsequently collected articles and patents along with recipients' professional careers for longitudinal analysis, the findings of this paper revealed that the 12 recipients with substantial contributions to science and technology had various characteristics in productivity and influence. Although age had no strong association with their research performance [53], not all extraordinary scientists had high research performance; these scientists were neither prolific nor influential. This is consistent with the finding of Chang et al. [13], who revealed that the research performance of 50 biological scientists with contributions to science that were recognized by earning the NMS and a fellowship from the American Academic of Arts and Sciences varied at the individual level. No clear relationship was observed between research performance and scientific contribution.

Although the diverse research performance of excellent scientists identified by previous studies may be parallel to our study's findings, we could not calculate the ratio of articles and patents for all of the scientists. Considering the difference in requirements between publishing and patenting, we anticipated that the ratios of articles to patents would be more than one for all 12 scientists, but we could not speculate the maximum of these ratios. The ratios of articles to patents for individual scientists that ranged from 0.03 to 28.33 indicated that we failed to predict some scientists' minimum productivity. In particular, unexpected findings include one recipient not having been granted U.S. patent and some recipients having publishing a relatively limited number of research articles indexed by the two large interdisciplinary databases, WoS and Scopus. This indicates the possibility that scientists' technological contributions may not be reflected in patents. This may also be true of the types of professional output that demonstrate scientific contribution. This type of scientist is not uncommon and not limited to the 12 recipients of the NMS and NMTI. For example, Nikola Tesla contributed to the design of the modern alternating current electricity supply system, but he did not publish scientific papers; however, he was recognized for his scientific contributions in the field of electrical engineering and mechanical engineering through his inventions [54].

Eleven recipients had published scholarly articles and held patented inventions. We further determined that the contributions for which they were awarded the NMS and NMTI were associated with their articles and patents. Excluding one subject (Achenbach) who did not hold a patent for an invention, all the subjects are now listed on the website of the National Inventors Hall of Fame. The website of the National Inventors Hall of Fame provided the specific U.S. patent number of their most influential patented inventions and their biographical information, which summarized the technological contributions that were consistent with contributions that led to them receiving the NMTI. Moreover, each scientist earned their NMS and NMTI from the same contribution or similar contributions. This indicates a blurring of the boundary between science and technology contributions. Moreover, for each recipient, some scientific articles and patents were topic related. The cumulative nature of intellectual endeavors means no single paper or patent fully represented their influence and contribution.

Compared with research productivity, research influence was revealed to have a stronger association with scientific contribution [13]. Although influence is not equivalent to contribution, these concepts are frequently confused because of the association between them. Therefore, in this study, citation-based counts were used to measure scientific and technological influence. The skewness of citation counts led to our expectation of few influential articles and patents, but we anticipated to observe at least one article and one patent produced by eminent scientists that were highly cited. In particular, the Matthew effect seems to be the natural consequence of several prestigious awards earned by the 12 scientists; these awards then boost their scientific and technological influence. The productivity and influence by year (Figs 2 and 3) and the years when the 12 scientists received their awards (Fig 1) indicated that some factors hamper identifying the association between the Matthew effect and citation counts received by year. One factor is that ever-increasing articles and patents enhance the likelihood of each scientist receiving citations. However, we could not determine the positive impact of awards on the increase in citation counts or the number of new citations received by scientists. Another factor is the decline in citation counts observed after some scientists received several awards. Despite 10 scientists publishing at least one top 0.1% highly cited article, two scientists did not publish such highly cited articles. Therefore, we could not identify substantial scientific contributions from only high citation levels.

Some awards also emphasize the importance of the research's impact on practice and society [55]. As scientists are also expected to contribute to society, such as through solving human problems, other types of influence associated with contributions, such as societal effects, have

started to be considered [56]. Altmetric scores are an emerging indicator for a type of societal influence other than the research influence, but they cannot be applied to scientists who published the majority of their papers before the emergence of Altmetrics. Measuring or presenting the social effect of individual researchers is still a challenge [55]. Therefore, although the 12 scientists have been recognized for their contribution to science and technology in terms of the NMS and NMTI, their contribution to society cannot be fully reflected in research performance measured by bibliometric indicators.

The various contributions resulting from discoveries cannot simply be measured through bibliometric indicators [57, 58]. Therefore, peer review mechanisms must remain the principal approach to assessing influences and contributions [59]. Furthermore, scientists capture different types of values, meeting their various needs for engagement in knowledge production and dissemination [20]. When societal contributions made by scientists are noticed, then some scientists are willing to further engage with society. However, the reality is that researchers' choice for knowledge dissemination is primarily affected by monetary incentives that are focused on the amount of productivity and influence that can be easily measured through bibliometric indicators. Therefore, other systems such as peer review must exist to measure scientists' contributions and achievements from other viewpoints.

On the basis of the current findings, we posit that scientists' contribution to technology may not be reflected in inventions (patents). The gap between bibliometric indicators and the peer review system for indicating scientific contribution was confirmed in eminent scientists with contribution to science and technology. To emphasize the differences in professional productivity and scholarly and technological influence, these scientists were divided into three types according to the values of the $h$-index for articles and patents. The type A scientist, Langer, exemplified a scientist with a substantial contribution to science and technology as evident in high productivity in articles and patents and high influence through the output. Moreover, Langer received numerous awards. The large differences in the $h$-index values for articles and patents between Langer and the other 11 scientists indicate that Langer cannot be considered a typical example of a scientist with exceptional contributions to humanity. Langer was an outlier compared with other scientists. In terms of the ratios of $h$-index (articles) to $h$-index (patents), Langer (type A), Holonyak (type B), Edgerton (type C), and Beckman (type C) had ratios close 1, ranging between 1.29 and 1.96. They had a higher degree of balance between publishing and patenting activities. However, only Langer and Holonyak balanced their publishing and patenting activities, demonstrating excellent research and technology performance. Most of the examined individuals were science- or technology-oriented scientists.

The principal limitation of this study was the small number of subjects. However, the scientists investigated in this study had obtained acknowledgment for exceptional contributions to science and technology. The NMS and NMTI were used as proxies. However, scientists who have made substantial contributions to science and technology are rare. We did not identify any other awards that could be used to increase the sample size. Despite the small sample size, we contributed to locating scientists with substantial contributions to science and technology, and we explored their research performance through investigating their articles and patents. We also examined the awards and honors that they had obtained to support the rationale for selecting these 12 scientists as subjects. Moreover, a longitudinal analysis reduced the effect of time. The results of this study demonstrate that scientists with substantial contributions may not be productive or influential from the perspective of bibliometric measures. Although studies have focused on the research performance of Nobel laureates and scientists that received other awards [60–62], the biggest contribution of this study is the observation that only a few eminent S&T scientists balanced their publishing and patents activities through high research performance in publishing and patenting. Scientists' contribution to science is not limited to

their research. Moreover, scientists' contribution to technology may not be evident from their inventions.

## Author Contributions

**Conceptualization:** Yu-Wei Chang, Dar-Zen Chen, Mu-Hsuan Huang.

**Data curation:** Yu-Wei Chang.

**Formal analysis:** Yu-Wei Chang.

**Funding acquisition:** Dar-Zen Chen, Mu-Hsuan Huang.

**Investigation:** Yu-Wei Chang.

**Methodology:** Yu-Wei Chang.

**Project administration:** Dar-Zen Chen.

**Resources:** Mu-Hsuan Huang.

**Supervision:** Mu-Hsuan Huang.

**Validation:** Dar-Zen Chen.

**Writing – original draft:** Yu-Wei Chang.

**Writing – review & editing:** Mu-Hsuan Huang.

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
