## [Decision Letter · Decision Letter 0]

18 Jun 2021

PONE-D-21-10702

Do extraordinary science and technology scientists balance their publishing and patenting activities?

PLOS ONE

Dear Dr. Huang,

Thank you for submitting your manuscript to PLOS ONE. After careful consideration, we feel that it has merit but does not fully meet PLOS ONE’s publication criteria as it currently stands. Therefore, we invite you to submit a revised version of the manuscript that addresses the points raised during the review process. I recommend also to follow the minor points raised by Reviewer #1.

The two reviewers are critical of your paper by agreeing on the necessity to better organize the Introduction and the Discussion and Conclusion sections. The figure need to be revised at least by introducing suitable and explicative caption. Please clarify the status of availability of the data. You declared that they are not available, and that "all relevant data are within the manuscript". Consider carefully the data availability policy adopted by PLOS ONE (https://journals.plos.org/plosone/s/data-availability). I recommend also to evaluate the minor point highlighted by reviewer #1.

We look forward to receiving your revised manuscript.

Kind regards,

Alberto Baccini, Ph.D.

Academic Editor

PLOS ONE

Journal Requirements:

2. Please consider changing the title so as to meet our title format requirement (https://journals.plos.org/plosone/s/submission-guidelines). In particular, the title should be "Specific, descriptive, concise, and comprehensible to readers outside the field" and in this case it is not informative and specific about your study's scope and methodology.

3.Please ensure that you include a title page within your main document. We do appreciate that you have a title page document uploaded as a separate file, however, as per our author guidelines (http://journals.plos.org/plosone/s/submission-guidelines#loc-title-page) we do require this to be part of the manuscript file itself and not uploaded separately.

Reviewers' comments:

Reviewer's Responses to Questions

**Comments to the Author**

1. Is the manuscript technically sound, and do the data support the conclusions?

Reviewer #1: Yes

Reviewer #2: No

2. Has the statistical analysis been performed appropriately and rigorously? 

Reviewer #1: Yes

Reviewer #2: Yes

3. Have the authors made all data underlying the findings in their manuscript fully available?

Reviewer #1: No

Reviewer #2: No

4. Is the manuscript presented in an intelligible fashion and written in standard English?

Reviewer #1: No

Reviewer #2: Yes

5. Review Comments to the Author

Reviewer #1: This study investigates the scientific and technological performance of the 12 scientists that were awarded both the NMS medal – an American prize recognizing scientific excellence – and the NMTI medal – an American prize given for outstanding technological contribution – in order to better understand the relationship between scientific publishing and patent activity of extraordinary scientists. The analysis of the bibliometric and patenting profiles of the 12 scientists shows that scientific and technological contributions that were awarded with prestigious prizes by the scientific community were not always matched by outstanding bibliometric and patenting performances. In this regard, the study is an interesting contribution to the debate on the capacity of bibliometric measures to capture “research excellence” and possibly to that on peer review vs. bibliometrics.

In my opinion, the study is a valuable contribution that deserves publication. However, I would suggest to the authors some revisions to improve clarity and readability of the paper.

First, I suggest the authors to better specify the focus of the paper and how it is related to the research questions. I was puzzled by the fact that the study mentions many different issues, including the relationship between science, technology, and academia (rows 15-27), the bibliometric and patenting profiles of outstanding scientists (rows 39-40), and if productivity, citation, and patent metrics are reliable indicators of scientific/technological excellence (rows 158-167), In my opinion, the main result of the paper is that these indicators are not reliable, as the outstanding scientists examined display highly diversified citation and patent profiles (rows 475-477, 531-532, and 552-553). I see that the two other topics are somehow related to this result, but I think that a more concise and clear presentation of the real focus of the paper would help the reader to follow the argument.

In particular, I suggest to better organize the Introduction and the Discussion and Conclusion sections, as some paragraphs seem to be scarcely connected with the main argument. For instance, the paragraph on Nobel laureates (rows 28-38) interrupts the presentation of the rationale of the study and it seems to me not relevant for the discussion, at least at this point of the paper. Similarly, the paragraph on the evaluation of societal effects (rows 520-530) seems to me scarcely related to the focus of the study.

The methodology applied is sound and well-tailored to the research questions. Quantitative analyses are well performed, and results support the paper’s claims. However, some figures may be improved:

Fig. 1: In the Lauterbur facet there is a blue line, does it mean something? If this is the case, please add the explanation.

Fig. 4 should be explained more extensively: what does the color code mean, for instance? What is “n” in the last row on the left? I suggest the author at least to expand the caption of this figure.

Fig. 5: it is not clear the rationale for the three types of scientists. Why were these ranges chosen to delimit type-B from type-C scientists? I suggest also to indicate the type by a color to improve the readability of the figure.

The authors cited mostly the appropriate literature, even if some key references that would perfectly fit in their argument are lacking. In particular, the sections on the relationship between citations and research quality should be integrated with a discussion of Aksnes et al. (2019) (DOI: 10.1177/2158244019829575) and similar contributions. Other references are suggested in the attached list of minor comments.

Lastly, I would suggest the authors get editing help from someone with full professional proficiency in English, as the meaning of some phrases and sentences in the manuscript is not always clear (see the list of minor comments attached).

Minor revisions and typos:

Row 19. “Notably” does not seem to me the right word. Maybe “However”?

Rows 28-38: I do not understand why this paragraph on Nobel laureates is placed here. It interrupts the argument. I would remove it.

Row 35: laureates_and

Row 42: “in sample <of> scientists”

Row 59: I am not sure that “assumption” is the right word here. Maybe “hypothesis” is better?

Row 89: Not sure “demonstrated” is the right word here. I suggest the authors to rephrase the entire sentence.

Row 127: “one scientific publication of a patent” should be “one scientific publication or a patent”.

Row 139-140: The meaning of this sentence is not clear. I suggest the authors to rephrase it.

Rows 143-145: I suggest the authors to integrate the discussion by commenting on Aksnes et al. (2019) (10.1177/2158244019829575), that is an in-depth review of the relationship between research quality and citations.

Row 162: I suggest the authors to add reference to Baccini and De Nicolao (2016) (10.1007/s11192-016-1929-y), a relevant study for this debate.

Row 209: I am not sure that the word “intelligence” is right here.

Row 229: “until” should be replaced with “before”, if a get the meaning of the sentence right.

Rows 231-232: It is not very clear how citations were calculated. Is it from the difference between the two?

Rows 287-288: Accumulation of prizes may be explained also by the Matthew Effect. I suggest the authors to briefly mention this alternative explanation.

Row 351: “he had been for a”

Row 483: I would replace “originate from” with “be reflected in” or “result in”.

Rows 485-490: The meaning of this paragraph is not clear to me. I suggest the authors to rephrase it and elaborate a bit on the relationship between inventions and discoveries to clarify the reasoning presented in this paragraph.

Ref n. 1: “National inVocation systems” should be changed in “National innovation systems”

Ref 25 “orreinforcement” should be replaced with “or reinforcement”

Ref. 32 I think it is better to cite e.g., MacRoberts and MacRoberts (2018) (10.1002/asi.23970)</of>

Reviewer #2: The paper summarizes the careers of 12 American scientists who might be considered as "excellent" performers both in science and technology. The authors try to understand what data about these scientists' publication and patenting activities can tell us about the career choices of successful scientists, and specifically about the relation between (academic) science and technology. Despite the small sample, the work is strictly based on a quantitative approach.

While I do appreciate the effort to better understand the sociology of science with from a scientometric perspective, I believe the paper suffers from three main problems.

First, the aims of the paper are unclear. The introduction (p. 4) spells out two very narrow and specific research questions (if these 12 people authored the same number of articles and patents, and if they are highly cited), and a very generic one (if their contributions are "demonstrated" by bibliometric indicators). The authors do not attempt to motivate why these questions are relevant. Then, the analysis only answers to the first two questions (see my point 2 concerning the third question), and the conclusions depart from all three questions, stating that the authors "have demonstrated that bibliometrics indicators are not a reliable measure of the value of discoveries in improving humanity" (pp. 23-24).

Second, the paper suffers from confusion in the use of terms and concepts. In some cases, this could be a language problem that the authors could possibly solve by asking a native speaker to proofread the manuscript. This is how I personally (and I myself am not a native speaker) interpret some odd statements such as "The research papers and patents produced ... served as proxies of scientific and technological intelligence" (pp. 9-10) or "The low number of winners indicates that this select group has been responsible for extraordinary contributions" (p. 3).

Other examples could be made, but my point is that quite obviously this is not just a language issue. Throughout the paper the authors use "contribution" and "influence" interchangeably (for example in pages 1, and 2), then clearly express the fact that they are not synonymous, but then e.g. on pag. 11 they write again that "citation counts were employed to represent [both] influence and contribution over time". Then for example on pag. 23 they deny this again ("Influence is frequently confused with contribution. In this study, only citation-based counts were used to measure and present scientific and technological influence"). To add to the confusion, sometimes they further add the notions of "performance" and "research originality" (beside the already recalled "intelligence", which I believe to be just a mistake). Perhaps, if the authors clarified better what they aim to measure or document, the statistical analysis too could be more in-depth and precise (see point below).

Third, the data analysis is rather superficial, and not geared at answering the third (larger) research question. For example, because of variability of citations across a same author's papers, the authors only consider the most highly cited paper for each author. Further, for years before 1970 the use citation data from Web of Science, and after 1970 from Scopus, without any consideration about how this might affect their results. The authors do not distinguish between citations received before and after the official recognition of the award(s) and therefore cannot tell us if impact (what they call "influence") is due to visibility and/or how is it affected by it.

More in general, I do not have precise ideas about how to demonstrate if "contribution" is measured by citation counts, so I am sorry I cannot offer constructive criticisms, but certainly the current analysis does not attempt to answer this question at all. Probably, totally different data would be required (including some measurable evidence of "contribution") and I would personally suggest to the authors to rather change the research question.

Concerning the empirical part, however, my biggest concern is that the authors draw general implications from a very (small and) select sample: both geographically (only US authors) and in terms of impact. They only write (p. 8) that this is an "appropriate" sample for a study of this kind, without much more reflection on this crucial methodological choice.

On the whole, I am sorry to say I believe these three main problems make the manuscript unsuitable for publication.

6. PLOS authors have the option to publish the peer review history of their article (what does this mean?). If published, this will include your full peer review and any attached files.

Reviewer #1: **Yes: **Eugenio Petrovich

Reviewer #2: **Yes: **Carlo D'Ippoliti

---

## [Author Response · Author response to Decision Letter 0]

24 Jul 2021

Thank you for the constructive comments on our manuscript. We have revised the paper based on the reviewer’ comments and recommendations. Revisions of the manuscript are highlighted in yellow. In addition, the point-by-point responses to the reviewer comments are appended below.

Reviewer #1: 

1. First, I suggest the authors to better specify the focus of the paper and how it is related to the research questions. I was puzzled by the fact that the study mentions many different issues, including the relationship between science, technology, and academia (rows 15-27), the bibliometric and patenting profiles of outstanding scientists (rows 39-40), and if productivity, citation, and patent metrics are reliable indicators of scientific/technological excellence (rows 158-167), In my opinion, the main result of the paper is that these indicators are not reliable, as the outstanding scientists examined display highly diversified citation and patent profiles (rows 475-477, 531-532, and 552-553). I see that the two other topics are somehow related to this result, but I think that a more concise and clear presentation of the real focus of the paper would help the reader to follow the argument. In particular, I suggest to better organize the Introduction and the Discussion and Conclusion sections, as some paragraphs seem to be scarcely connected with the main argument. For instance, the paragraph on Nobel laureates (rows 28-38) interrupts the presentation of the rationale of the study and it seems to me not relevant for the discussion, at least at this point of the paper. Similarly, the paragraph on the evaluation of societal effects (rows 520-530) seems to me scarcely related to the focus of the study.

Response:

 Thank you for your constructive suggestions and comments. We have reorganized and revised the Introduction and the Discussion and Conclusions sections. In the Introduction section, we have revised the third research question and have deleted disconnected sentences including original lines 15–40 to clarify and strengthen statements and arguments. Regarding the Discussion and Conclusions section, in addition to deleting the irrelevant statements that you mentioned, we have added new references and made considerable revisions to improve and clarify the statements related to the discussion and conclusion. 

2. Fig. 1: In the Lauterbur facet there is a blue line, does it mean something? If this is the case, please add the explanation.

Response:

 We have added an explanation for the blue vertical line in two scientists’ data to the description of Fig. 1. The blue vertical line indicated the year when the scientist was awarded the Nobel Prize.

3. Fig. 4 should be explained more extensively: what does the color code mean, for instance? What is “n” in the last row on the left? I suggest the author at least to expand the caption of this figure.

Response: 

 We have revised Fig 4 to enhance the visual effect and reader comprehension. More details on Fig 4 have been added to the corresponding paragraph. In addition, we have expanded the caption of Fig 4. 

4. Fig. 5: it is not clear the rationale for the three types of scientists. Why were these ranges chosen to delimit type-B from type-C scientists? I suggest also to indicate the type by a color to improve the readability of the figure.

Response:

 We have added statements to clarify the rationale for three types of scientists. We have also revised Fig 5 to improve its visual effect, including the use of colors for scientists of different types.

5. The authors cited mostly the appropriate literature, even if some key references that would perfectly fit in their argument are lacking. In particular, the sections on the relationship between citations and research quality should be integrated with a discussion of Aksnes et al. (2019) (DOI: 10.1177/2158244019829575) and similar contributions. Other references are suggested in the attached list of minor comments.

Response:

 We have added the three articles that you suggested and other references to the list of references and incorporated them into the statements and discussions related to citations, research quality, influence, and contributions.

6. Lastly, I would suggest the authors get editing help from someone with full professional proficiency in English, as the meaning of some phrases and sentences in the manuscript is not always clear (see the list of minor comments attached). 

Response: 

 The manuscript has undergone professional editing.

7. Minor revisions and typos:

Row 19. “Notably” does not seem to me the right word. Maybe “However”?

Response:

 We have replaced “Notably” with “However.”

Rows 28-38: I do not understand why this paragraph on Nobel laureates is placed here. It interrupts the argument. I would remove it.

Response:

 We have deleted this paragraph.

Row 42: “in sample scientists” 

Response:

 We have deleted the word “sample.”

Row 59: I am not sure that “assumption” is the right word here. Maybe “hypothesis” is better?

Response:

 We have replaced “assumption” with “hypothesis.”

Row 89: Not sure “demonstrated” is the right word here. I suggest the authors to rephrase the entire sentence.

Response: 

 We have revised the third research question.

Row 139-140: The meaning of this sentence is not clear. I suggest the authors to rephrase it.

Response:

 We have rephrased this sentence.

Rows 143-145: I suggest the authors to integrate the discussion by commenting on Aksnes et al. (2019) (10.1177/2158244019829575), that is an in-depth review of the relationship between research quality and citations.

Row 162: I suggest the authors to add reference to Baccini and De Nicolao (2016) (10.1007/s11192-016-1929-y), a relevant study for this debate.

Ref. 32 I think it is better to cite e.g., MacRoberts and MacRoberts (2018) (10.1002/asi.23970)

Response:

 We have added the three references as suggested and integrated them into the Literature Review and the Discussion and Conclusions sections.

Row 209: I am not sure that the word “intelligence” is right here.

Response:

 We have replaced “intelligence” with “output.”

Row 229: “until” should be replaced with “before”, if a get the meaning of the sentence right.

Response:

 We have replaced “until” with “before.”

Rows 231-232: It is not very clear how citations were calculated. Is it from the difference between the two?

Response:

 We have rephrased this sentence to enhance clarity.

Rows 287-288: Accumulation of prizes may be explained also by the Matthew Effect. I suggest the authors to briefly mention this alternative explanation.

Response:

 The relevance of the Matthew effect to our study has been addressed in the Discussion section.

Row 35: laureates_and

Row 127: “one scientific publication of a patent” should be “one scientific publication or a patent”.

Row 351: “he had been for a”

Ref 25 “orreinforcement” should be replaced with “or reinforcement”

Response: 

 We have examined the whole paper and corrected all typos.

Row 483: I would replace “originate from” with “be reflected in” or “result in”.

Response:

 We have replaced “originate from” with “be reflected in.”

Rows 485-490: The meaning of this paragraph is not clear to me. I suggest the authors to rephrase it and elaborate a bit on the relationship between inventions and discoveries to clarify the reasoning presented in this paragraph.

Response:

 We have rephrased the relevant text in the Discussion and Conclusions section.

Ref n. 1: “National inVocation systems” should be changed in “National innovation systems”

Response:

 We have deleted this reference.

Reviewer #2: 

1. First, the aims of the paper are unclear. The introduction (p. 4) spells out two very narrow and specific research questions (if these 12 people authored the same number of articles and patents, and if they are highly cited), and a very generic one (if their contributions are "demonstrated" by bibliometric indicators). The authors do not attempt to motivate why these questions are relevant. Then, the analysis only answers to the first two questions (see my point 2 concerning the third question), and the conclusions depart from all three questions, stating that the authors "have demonstrated that bibliometrics indicators are not a reliable measure of the value of discoveries in improving humanity" (pp. 23-24).

Response:

 Thank you for your comments and suggestions. We have rephrased and reorganized the Introduction section to clarify the purpose and research questions of this paper. In particular, we have revised the third research questions and strengthened statements related to the three research questions to enhance the relevance of these statements. In addition, we have revised the Discussion and Conclusions section to ensure the discussions are connected to the research questions. 

2. Second, the paper suffers from confusion in the use of terms and concepts. In some cases, this could be a language problem that the authors could possibly solve by asking a native speaker to proofread the manuscript. This is how I personally (and I myself am not a native speaker) interpret some odd statements such as "The research papers and patents produced ... served as proxies of scientific and technological intelligence" (pp. 9-10) or "The low number of winners indicates that this select group has been responsible for extraordinary contributions" (p. 3). Other examples could be made, but my point is that quite obviously this is not just a language issue. 

Throughout the paper the authors use "contribution" and "influence" interchangeably (for example in pages 1, and 2), then clearly express the fact that they are not synonymous, but then e.g. on pag. 11 they write again that "citation counts were employed to represent [both] influence and contribution over time". Then for example on pag. 23 they deny this again ("Influence is frequently confused with contribution. In this study, only citation-based counts were used to measure and present scientific and technological influence"). To add to the confusion, sometimes they further add the notions of "performance" and "research originality" (beside the already recalled "intelligence", which I believe to be just a mistake). Perhaps, if the authors clarified better what they aim to measure or document, the statistical analysis too could be more in-depth and precise (see point below).

Response:

 We have used an English editing service to improve our writing quality. Regarding the meaning of some concepts, such as influence, contribution, performance, and research originality, we have examined the full paper and revised related statements. In particular, we have strengthened the rationale for and what we analyzed through our research questions.

3. Third, the data analysis is rather superficial, and not geared at answering the third (larger) research question. For example, because of variability of citations across a same author's papers, the authors only consider the most highly cited paper for each author.

Response:

 We have revised the third research questions and enhanced the connection between the three research questions.

4. Further, for years before 1970 the use citation data from Web of Science, and after 1970 from Scopus, without any consideration about how this might affect their results. 

Response:

 We have provided further information regarding the citation data collected from Web of Science (WoS) and Scopus. Scopus covers more journals than does WoS. This helped us collect the maximum number of articles published by the 12 scientists. However, Scopus does not provide data on the annual number of citations received by articles before 1970. Thus, for articles published before 1970 and that have received citations before 1970, their total number of citations would be under-counted. To address this problem, we verified whether articles published before 1970 were indexed by WoS and had annual citation count data before 1970. Therefore, for articles published before 1970, their annual citation counts were collected from WoS (before 1970) and Scopus (after 1970).

5. The authors do not distinguish between citations received before and after the official recognition of the award(s) and therefore cannot tell us if impact (what they call "influence") is due to visibility and/or how is it affected by it. 

Response:

 We have recorded citation counts received by year and illustrated the data in Fig 3. In addition, Fig 1 presents the years when each scientist received awards. With reference to Figs 1 and 3, we can observe the changes in citation counts before and after awards. Statements related to citation counts and awards have been added to the Discussion and Conclusions section.

6. More in general, I do not have precise ideas about how to demonstrate if "contribution" is measured by citation counts, so I am sorry I cannot offer constructive criticisms, but certainly the current analysis does not attempt to answer this question at all. Probably, totally different data would be required (including some measurable evidence of "contribution") and I would personally suggest to the authors to rather change the research question.

Response:

 We have expanded and clarified statements related to citation counts, influence, and contribution as well as our research motives, and we have revised the third research question. As mentioned in our responses to questions 1–3, the description of what we measured for each research question has been expanded and clarified. 

7. Concerning the empirical part, however, my biggest concern is that the authors draw general implications from a very (small and) select sample: both geographically (only US authors) and in terms of impact. They only write (p. 8) that this is an "appropriate" sample for a study of this kind, without much more reflection on this crucial methodological choice.

Response:

 Regarding your biggest concern, we have explained why only 12 subjects were targeted in the study in our Methodology section. To identify scientists with substantial contributions to science and technology, we considered those that had received two specific distinguished awards, with one award emphasizing scientific contribution and the other focusing on technological contribution. We have expanded and clarified our rationale and method used for targeting the 12 scientists.

---

## [Decision Letter · Decision Letter 1]

7 Sep 2021

PONE-D-21-10702R1

Do extraordinary science and technology scientists balance their publishing and patenting activities?

PLOS ONE

Dear Dr. Huang,

Thank you for submitting your manuscript to PLOS ONE. After careful consideration, we feel that it has merit but does not fully meet PLOS ONE’s publication criteria as it currently stands. Therefore, we invite you to submit a revised version of the manuscript that addresses the points raised during the review process.

I think that the paper can be accepted for publication,  after you have carried out the corrections as suggested by the two reviewer. Please consider carefully the minor corrections indicated by both reviewers, and feel free to simplify the introduction according to the comment of the first reviewer. 

We look forward to receiving your revised manuscript.

Kind regards,

Alberto Baccini, Ph.D.

Academic Editor

PLOS ONE

Journal Requirements:

1. Please change the title so as to meet our title format requirement (https://journals.plos.org/plosone/s/submission-guidelines). In particular, the title should be "Specific, descriptive, concise, and comprehensible to readers outside the field" and in this case it is not informative and specific about your study's scope, methodology, and findings.

Additional Editor Comments (if provided):

Reviewers' comments:

Reviewer's Responses to Questions

**Comments to the Author**

1. If the authors have adequately addressed your comments raised in a previous round of review and you feel that this manuscript is now acceptable for publication, you may indicate that here to bypass the “Comments to the Author” section, enter your conflict of interest statement in the “Confidential to Editor” section, and submit your "Accept" recommendation.

Reviewer #1: (No Response)

Reviewer #2: All comments have been addressed

2. Is the manuscript technically sound, and do the data support the conclusions?

Reviewer #1: Yes

Reviewer #2: Yes

3. Has the statistical analysis been performed appropriately and rigorously? 

Reviewer #1: Yes

Reviewer #2: Yes

4. Have the authors made all data underlying the findings in their manuscript fully available?

Reviewer #1: Yes

Reviewer #2: (No Response)

5. Is the manuscript presented in an intelligible fashion and written in standard English?

Reviewer #1: Yes

Reviewer #2: Yes

6. Review Comments to the Author

Reviewer #1: The new version of the manuscript is clearer and better organized compared to the first one. The paper’s focus is now clearly explained, and the research questions plainly spelled out.

However, I still have some troubles with the Introduction, which, in its present form, is too long, with paragraphs that are not always mutually well connected. My suggestion to the authors is to shorten it and remove several details that are already provided in the following sections of the papers (e.g., the debate on the reliability of citation counts for measuring influence is better presented in the Literature Review section, the history of the NMS and NMTI can be left to the Methodology section without weighing the Introduction down, and so on). I would suggest the authors to limit the Introduction to the presentation of the rationale and relevance of their research questions, leaving the context and other details to the Literature Review section.

Some repetitions may also be reduced to enhance readability throughout the paper (e.g., the fact that filing patents is more challenging than publishing articles is repeated at least three times in different points of the paper).

Lastly, there are still few typos that should be corrected:

Abstract: row 30: there is a missing “s” after “scientist”.

Row 86: it seems the verb is missing in this sentence.

Row 249: “scientist” should be turned to plural.

Reviewer #2: The authors addressed the most important concerns I and the other referees had raised. There remains a difference in points of views, which should not prevent publication of the manuscript, of course.

A final, minor remark: please clarify and streamline throughout the date(s) at which the data are updated. It seems to me, 2018 (for sample selection) and 2019 (for citations data), but this should be more clearly reported. Please note that on pag. 11 (row 252) you write "as of 2004": what does that mean, is that a typo?

7. PLOS authors have the option to publish the peer review history of their article (what does this mean?). If published, this will include your full peer review and any attached files.

Reviewer #1: **Yes: **Eugenio Petrovich

Reviewer #2: **Yes: **Carlo D'Ippoliti

---

## [Author Response · Author response to Decision Letter 1]

8 Oct 2021

We appreciate the time and efforts that the reviewers have dedicated to providing valuable feedback on our manuscript. We have revised the paper based on the reviewers’ comments. The revisions are highlighted in yellow in the manuscript. Here is a point-by-point response to the reviewers’ comments and concerns.

Reviewer #1: 

Comment 1: The new version of the manuscript is clearer and better organized compared to the first one. The paper’s focus is now clearly explained, and the research questions plainly spelled out. However, I still have some troubles with the Introduction, which, in its present form, is too long, with paragraphs that are not always mutually well connected. My suggestion to the authors is to shorten it and remove several details that are already provided in the following sections of the papers (e.g., the debate on the reliability of citation counts for measuring influence is better presented in the Literature Review section, the history of the NMS and NMTI can be left to the Methodology section without weighing the Introduction down, and so on). I would suggest the authors to limit the Introduction to the presentation of the rationale and relevance of their research questions, leaving the context and other details to the Literature Review section. 

Response: We agree with this comment, and we have shortened the Introduction section by moving some statements to other sections, leaving the Introduction section clearer and more precise.

Comment 2: Some repetitions may also be reduced to enhance readability throughout the paper (e.g., the fact that filing patents is more challenging than publishing articles is repeated at least three times in different points of the paper).

Response: We have deleted the repetitive statements that you pointed out. And we have examined the whole paper and deleted some other repetitions to enhance readability, thanks for your suggestion.

Comment 3: There are still few typos that should be corrected: Abstract: row 30: there is a missing “s” after “scientist”. Row 86: it seems the verb is missing in this sentence. Row 249: “scientist” should be turned to plural.

Response: We have revised the typos that you pointed out and have also checked the whole paper to correct the other typos.

Reviewer #2: 

Comment 1: The authors addressed the most important concerns I and the other referees had raised. There remains a difference in points of views, which should not prevent publication of the manuscript, of course. A final, minor remark: please clarify and streamline throughout the date(s) at which the data are updated. It seems to me, 2018 (for sample selection) and 2019 (for citations data), but this should be more clearly reported. Please note that on pag. 11 (row 252) you write "as of 2004": what does that mean, is that a typo?

Response: We have revised the typo that you identified on page 11, which is supposed to be “as of 2019”, thank you for pointing this out. We have also examined all the dates in this paper and made sure that they are correct.

We have indicated that the latest year of the published papers and patents that we used for analysis is 2018. The bibliographic records of the papers, which include the number of citations for each paper, were downloaded on January 29, 2019. Because the cumulative number of citations received by each paper was updated on a date close to the end of 2018, the number of citations received by each paper closely reflects their influence by the end of 2018.

For the patents, we have also elaborated on the relevant statements to clarify the dates. The bibliographic records of the patents were collected in February 2019. Because the number of patent-related citations is not provided in the USPTO database, we calculated the number of citations received on the basis of the references for all patents. The list of references does not change after the patents are granted. Therefore, 2018 is the latest year when the number of patent citations is recorded.

---

## [Editor Report · Decision Letter 2]

20 Oct 2021

Do extraordinary science and technology scientists balance their publishing and patenting activities?

PONE-D-21-10702R2

Dear Dr. Huang,

We’re pleased to inform you that your manuscript has been judged scientifically suitable for publication and will be formally accepted for publication once it meets all outstanding technical requirements.

Kind regards,

Alberto Baccini, Ph.D.

Academic Editor

PLOS ONE
---

## [Editor Report · Acceptance letter]

25 Oct 2021

PONE-D-21-10702R2 

Do extraordinary science and technology scientists balance their publishing and patenting activities? 

Dear Dr. Huang:

I'm pleased to inform you that your manuscript has been deemed suitable for publication in PLOS ONE. Congratulations! Your manuscript is now with our production department. 

Kind regards, 

on behalf of

Prof. Alberto Baccini 

Academic Editor

PLOS ONE